# Human MIKO-1, a Hybrid Protein That Regulates Macrophage Function, Suppresses Lung Fibrosis in a Mouse Model of Bleomycin-Induced Interstitial Lung Disease

**DOI:** 10.3390/ijms23179669

**Published:** 2022-08-26

**Authors:** Takuya Kotani, Masaki Ikemoto, Shogo Matsuda, Ryota Masutani, Tohru Takeuchi

**Affiliations:** 1Department of Internal Medicine (IV), Osaka Medical and Pharmaceutical University, Takatsuki 569-8686, Japan; 2Division of Central Laboratory, Osaka Medical and Pharmaceutical University, Osaka 569-8686, Japan

**Keywords:** S100, macrophages, pulmonary fibrosis, interstitial lung disease, connective tissue disease

## Abstract

Although interstitial lung disease (ILD) is a life-threatening pathological condition that causes respiratory failure, the efficiency of current therapies is limited. This study aimed to investigate the effects of human MIKO-1 (hMIKO-1), a hybrid protein that suppresses the abnormal activation of macrophages, on murine macrophage function and its therapeutic effect in a mouse model of bleomycin-induced ILD (BLM-ILD). To this end, the phenotype of thioglycolate-induced murine peritoneal macrophages co-cultured with hMIKO-1 was examined. The mice were assigned to normal, BLM-alone, or BLM + hMIKO-1 groups, and hMIKO-1 (0.1 mg/mouse) was administered intraperitoneally from day 0 to 14. The mice were sacrificed on day 28, and their lungs were evaluated by histological examination, collagen content, and gene expression levels. hMIKO-1 suppressed the polarization of murine macrophages to M2 predominance in vitro. The fibrosis score of lung pathology and lung collagen content of the BLM + hMIKO-1 group were significantly lower than those in the BLM-alone group. The expression levels of *TNF-**α, IL-6, IL-1**β, F4/80*, and *TIMP-1* in the lungs of the BLM + hMIKO-1 group were significantly lower than those in the BLM-alone group. These findings indicate that hMIKO-1 reduces lung fibrosis and may be a future therapeutic candidate for ILD treatment.

## 1. Introduction

Interstitial lung disease (ILD) is a life-threatening pathological condition that can cause respiratory failure [1]. There are several types of ILD with varying characteristics. Idiopathic pulmonary fibrosis (IPF) is mainly caused by progressive fibrosis in the pulmonary interstitium, and the inflammatory mechanisms are a minor component of the pathogenesis [2]. In ILD associated with connective tissue disease (CTD), the pathophysiology of inflammation and fibrosis in the lung interstitium is mixed [3,4]. Recently, coronavirus disease (COVID-19) infections have been rampant worldwide. Pneumonia caused by COVID-19 presents as ILD due to abnormal activation of innate immunity, including macrophages, and the sequelae of irreversible lung fibrosis have become a clinical problem [5]. Although corticosteroids and immunosuppressants are used for controlling the inflammatory pathology of ILD [6], there are many cases wherein ILD progression cannot be suppressed. Furthermore, the long-term use of immunosuppressive therapy causes infections and drug-induced side effects. Pirfenidone and nintedanib have been used to suppress the progression of fibrosis; however, their effects are limited [7]. Therefore, there is a need to develop better ILD treatments in terms of efficacy and safety.

S100 is a calcium-binding protein found in the synovial fluid of patients with rheumatoid arthritis and is produced by myeloid immune cells, such as macrophages and neutrophils [8,9]. Among the S100 proteins, S100A8 and S100A9 exist mainly as heterodimers (S100A8/A9; calprotectin) in vivo. S100A8/A9 was reported as a candidate inflammatory biomarker in rheumatoid arthritis and ulcerative colitis (UC) [10,11,12]. In ILD, serum S100A8/A9 levels are elevated in dermatomyositis (DM) patients with ILD compared with those in normal controls and are associated with ILD progression [13]. However, the functions of S100A8 and S100A9 have not yet been clarified.

We recently detected an autocrine signal transduction pathway involving rat S100A8 (rS100A8) and rS100A9 in macrophages and identified a cluster of differentiation 68 (CD68) on the macrophages of rats as a receptor-like protein for rS100A8 with or without rS100A9 in the signal transduction pathway of these cells, with sugar chains on these molecules being closely involved in signal transduction [14,15,16]. Next, we created a new hybrid protein (rat MIKO-1; rMIKO-1) by referring to the amino acid (AA) sequences of rS100A8 and rS100A9. rMIKO-1 suppressed the abnormal activation of LPS-activated rat peritoneal macrophages in an in vitro system and significantly suppressed enteritis in UC model rats [17].

Activated macrophages play a central role in the pathophysiology of ILD. In the process of pulmonary fibrosis during ILD, chemokines secreted from activated macrophages induce the transformation of fibroblasts into myofibroblasts, and the mechanism of overproduction of extracellular matrix was reported [18,19,20]. In light of this information, we tested the hypothesis that MIKO-1, which suppresses abnormal activation of macrophages, has a favorable effect on ILD. To this end, we investigated the effects of human MIKO-1 (hMIKO-1) on mouse peritoneal macrophages and its therapeutic effects in a mouse model of ILD.

## 2. Results

### 2.1. hMIKO-1 Effectively Reduced Fibrosis in ILD Lung

An increased collagen deposition caused fibrosis in the BLM-ILD lungs, and hMIKO-1 administration reduced this fibrosis (Figure 1A). The fibrosis score was significantly higher in the BLM-alone group than in the normal group but significantly lower in the BLM + hMIKO-1 group than in the BLM-alone group (Figure 1B). Collagen content was significantly higher in the BLM-alone group than in the normal group but significantly lower in the BLM + hMIKO-1 group than in the BLM-alone group (Figure 1C).

### 2.2. hMIKO-1 Regulated the Genes Associated with the Anti-Inflammatory and Anti-Fibrotic Effects in ILD

In order to evaluate the anti-inflammatory and anti-fibrotic effects of hMIKO-1, whole lung mRNA expression of inflammatory cytokines, matrix metalloproteinase-9 (MMP-9), and tissue inhibitor of metalloproteinase-1 (TIMP-1) were analyzed using quantitative reverse transcription-PCR (qRT-PCR). The relative mRNA expression levels of tumor necrosis factor alpha (TNF-α) (Figure 2A), interleukin (IL)-6 (Figure 2B), IL-1β (Figure 2C), F4/80 (Figure 2D), and TIMP-1 (Figure 2E) were significantly increased in the BLM-alone group compared with those in the normal group and significantly reduced in the BLM + hMIKO-1 group compared with those in the BLM-alone group. Although the relative mRNA expression of MMP-9 was significantly reduced in the BLM-alone group compared with that in the normal group, there was no significant difference between the BLM-alone and BLM + hMIKO-1 groups (Figure 2F). The MMP-9/TIMP-1 ratio was significantly lower in the BLM-alone group than in the normal group (Figure 2G). Although the MMP-9/TIMP-1 ratio was not significantly different between the BLM + hMIKO-1 and BLM-alone groups, it tended to increase (*p =* 0.097).

### 2.3. hMIKO-1 Suppressed the Polarity of Mouse Macrophages to M2 Dominance In Vitro

The results of the examination of cell surface antigens of murine macrophages are shown in Figure 3. The expression of CD64 was significantly increased, while that of CD163 was significantly reduced in the hMIKO-1 co-culture group compared with that of the control group in which hMIKO-1 was not added. The CD163/CD64 ratio was significantly lower in the hMIKO-1 co-cultured group than in the control group. Regarding cytokine concentrations in the culture supernatant of murine macrophages, IL-12 and IL-10 levels were significantly higher in the hMIKO-1 co-culture group than in the control group (Figure 4A), and the IL-10/IL-12 ratio was significantly lower (Figure 4B). Gene expression analysis of murine macrophages revealed that CD64, IL-12, and IL-10 levels were significantly increased in the hMIKO-1 co-culture group compared with those in the control group, but there was no difference in CD163 expression (Figure 4C). The CD163/CD64 ratio was significantly lower in the hMIKO-1 co-culture group than in the control group (Figure 4D). These results indicated that co-culture with hMIKO-1 suppressed the polarization of murine macrophages to M2 predominance.

## 3. Discussion

In the present study, hMIKO-1 effectively reduced lung fibrosis and regulated the expression of genes related to anti-inflammatory and anti-fibrotic effects in BLM-ILD mice. hMIKO-1 suppressed the polarization of murine macrophages to M2 predominance in vitro.

In the early acute inflammatory process in BLM-ILD mice, lung macrophages are directly and indirectly involved in lung injury by producing MMP and inflammatory cytokines such as TNF-α, IL-6, and IL-1β [21,22]. During the chronic inflammatory phase, M2 macrophages infiltrate the alveoli and induce continuous fibroblast differentiation and proliferation through the production of fibrosis-promoting mediators such as transforming growth factor-β1 and platelet-derived growth factor [18,19]. In this study, hMIKO-1 administration suppressed pulmonary fibrosis and the expression of F4/80, inflammatory cytokines (TNF-α, IL-6, and IL-1β), and fibrotic factors in the lungs of BLM-ILD mice. The results of our in vivo experiments suggest that hMIKO-1 may suppress lung inflammation and secondary lung fibrosis by suppressing macrophages in the lung lesions. Additionally, since M2 macrophages are mainly involved in the process of lung fibrosis in BLM-ILD mice [18,19,20] and hMIKO-1 suppressed the M2 polarization of macrophages in the in vitro experiments in this study, suppression of macrophage M2 polarization by hMIKO-1 was also considered one of the mechanisms responsible for suppressing lung fibrosis in BLM-ILD mice.

In patients with IPF, M2 macrophages excessively infiltrate the alveoli, causing fibrosis during the repair process of lung damage [19,22]. In CTD-ILD, CD163+ macrophages infiltrate the alveoli of patients with DM-ILD, which is associated with the severity of ILD [23]. We previously reported that in lung pathology of ILD in patients with microscopic polyangiitis, chemokine C-C motif ligand 2 (CCL2)-producing CD68+/CD163+ macrophages infiltrate the alveoli, and serum CCL2 levels significantly correlate with the progression of lung fibrosis [24]. Although the degree may vary depending on the background disease, it is considered that M2 macrophages infiltrate the alveoli in ILD and induce lung fibrosis during inflammation repair. Thus, hMIKO-1, which suppresses the M2 polarization of macrophages, may be a potential therapeutic target for ILD.

Thioglycolate-induced murine peritoneal macrophages are prone to M2 polarization [25]. In this study, thioglycolate-induced mouse peritoneal macrophages were significantly more likely to be polarized to M2 by cell surface antigens and culture supernatants (Appendix A), similarly to that in previous studies. In this study, hMIKO-1 suppressed the M2 polarization by significantly increasing CD64 expression and reducing CD163 expression in mouse peritoneal macrophage surface antigens. In addition, although hMIKO-1 increased the production of IL-12 and IL-10 in mouse peritoneal macrophages, the IL-10/IL-12 ratio was significantly reduced. This result supported the suppression of M2 polarization by hMIKO-1. The gene expression in murine peritoneal macrophages also supported this phenomenon.

There are several limitations to this study. Suppression of macrophages by hMIKO-1 was speculated to be one of the mechanisms of efficacy in BLM-ILD mice; however, due to the hypothesis inferred from in vitro experiments, the detailed mechanism of action in vivo is a future research topic. The properties of thioglycolate-induced peritoneal macrophages and those associated with lung lesions may differ in BLM-ILD mice. hMIKO-1 may also indirectly affect other inflammatory cells, such as T and B cells. Since macrophages are involved not only in autoimmune responses but also in infection defense, continuous exploration of the effects of hMIKO-1 on the immune system is necessary.

## 4. Materials and Methods

### 4.1. Ethics

The Institutional Animal Care and Use Committee of Osaka Medical and Pharmaceutical University approved all research protocols (approval ID: 21050-A), including surgical procedures and animal care. All experiments were performed in accordance with relevant guidelines and regulations.

### 4.2. Human MIKO-1

We designed a new hybrid protein (hMIKO-1) by referring to the AA sequences of hS100A8 and hS100A9 [8,14,15,16] and artificially synthesized cDNA for the hybrid protein using gene technology (Figure 5). The entire frame of hMIKO-1 protein was schematically indicated by a horizontal bar. The hMIKO-1 frame was as follows: 21 AA residues from the C-terminus of hS100A9 were added to the C-terminus of hS100A8.

### 4.3. Animal Models and Surgical Procedure

Female 13-week-old C57BL/6J mice (Shimizu Laboratory Supplies, Kyoto, Japan) were anesthetized with an intraperitoneal injection of 5% isoflurane for 3 min. The mice were then maintained on 1.5–2.0% isoflurane and divided into three groups. These included untreated mice (normal), mice with bleomycin (BLM)-induced ILD (BLM-alone), and mice with BLM-induced ILD plus hMIKO-1 administration (BLM + hMIKO-1). All experiments were performed on ten mice per group. The BLM solution was prepared by mixing sterile BLM sulfate powder (Nippon Kayaku, Tokyo, Japan) with sterile normal saline. A dose of 3 mg BLM in a total volume of 100 μL sterile saline was injected subcutaneously using an osmotic minipump (Alzet; DURECT, Cupertino, CA, USA) from day 0 to day 7, as previously described [26]. hMIKO-1 (0.1 mg/50 mM phosphate buffer (pH 7.4)/0.9% NaCl sterilized (0.2 mL/mouse) was administered intraperitoneally from day 0 to day 14. Control mice received the same volume (0.2 mL) of 50 mM phosphate buffer (pH 7.4)/0.9% NaCl sterilized (buffer A) alone. Macrophages were obtained from the abdominal cavities of normal mice. The mice were anesthetized with 5% isoflurane for 1 min and euthanized by cervical dislocation on day 21 (total: 28 days); their lungs were then harvested for analysis.

### 4.4. Preparation of Mouse Macrophages and Co-Culture with hMIKO-1

Murine macrophages were elicited by intraperitoneal injection of 1 mL sterilized 4% thioglycolate (Sigma, St. Louis, MO, USA) in buffer A. After 4 days, the mice were sacrificed, and the abdominal cavities were flushed with 5 mL ice-cold buffer A to harvest the cells. After centrifugation at 1000 rpm and 4 °C for 10 min, the cells were resuspended in an RPMI-1640 medium supplemented with 10% fetal bovine serum (medium A).

In our previous study, the concentration of hMIKO-1 that suppressed the abnormal activation of LPS-treated rat macrophages in the in vitro experiments was used as a reference [17]. The cells were seeded onto 6-well culture plates, and each sample (3 × 10^5^ cells per well) was cultivated with or without hMIKO-1 (approximately 25 μg/mL) for 24 h at 37 °C in medium A. After incubation, non-adherent cells were removed, and culture supernatants were stored at −30 °C to measure inflammatory cytokine concentration. Adherent cells were washed twice with buffer A and then collected in 1 mL of medium A using a scraper. The cell suspension was slightly centrifuged at 112× *g* and 4 °C for 10 min, and the supernatant was discarded.

### 4.5. Flow Cytometry

In order to confirm the alterations in the levels of cell surface antigens between the control and hMIKO-1 co-cultured groups, the cells were washed with PBS and then incubated with 10 μL PE-cyanine7-conjugated mouse anti-mouse monoclonal CD64 antibody (BioLegend, San Diego, CA, USA) or APC-conjugated rat anti-mouse monoclonal CD163 antibody (BioLegend, San Diego, CA, USA) for 30 min at 4 °C in the dark. After incubation, the samples were washed and analyzed via flow cytometry using a Navios cytometer (Beckman Coulter, Brea, CA, USA).

### 4.6. Enzyme-Linked Immunosorbent Assays

Supernatants were evaluated for interleukin-12 (IL-12) and IL-10 levels using a Mouse IL-10 ELISA Kit (Proteintech Group, Rosemont, IL, USA) and an LBIS Mouse IL-12 ELISA Kit (FUJIFILM Wako Shibayagi, Gunma, Japan). The experiments were independently repeated four times for each sample and performed in duplicate.

### 4.7. In Vitro Analysis of Murine Macrophages via qRT-PCR

Total RNA was extracted from cultured murine macrophages. Following RNA extraction using the RNeasy Mini Kit (Qiagen Ltd., Manchester, UK), cDNA was synthesized using an ExScript RT kit (Takara, Shiga, Japan), and gene amplification was performed using an ABI PRISM 7000 Sequence Detection System (Applied Biosystems, Tokyo, Japan) according to the manufacturer’s instructions. The primer sequences for *CD64, CD163, IL-12, IL-10*, and *GAPDH* (housekeeping gene) are summarized in Appendix A. The relative mRNA expression of each target gene was calculated using the comparative C_T_ method. The experiments were independently repeated four times for each sample and performed in triplicate.

### 4.8. Histological Evaluation and Assessment of the Fibrotic Area

The right middle lobe of the lung of each BLM-ILD mouse was fixed in 4% PFA/PBS for 6 h, followed by overnight incubation in 20% sucrose/PBS. The tissues were embedded in paraffin, cut into 5 μm sections, and stained with Masson’s trichrome stain. For each slide, the five most severely injured non-overlapping fields (magnification 200×) of the lung parenchyma were evaluated as previously described [27]. Lung fibrosis was quantified in the histological specimens using a numerical scale. The severity of fibrotic changes in each observed microscopic field of a given lung section was assessed and scored using a modified Ashcroft scale from 0 to 8 [28]. The overall severity of each lung section is expressed as the mean of the scores of the observed microscopic fields. Histological examinations were performed by three independent observers.

### 4.9. Hydroxyproline Assay

Next, hydroxyproline content was determined in the lung tissue. The left lung of each mouse was harvested and stored at −70 °C. Each sample was treated with 0.5 mol/L acetic acid containing pepsin (0.3 mg/10 mg tissue) at 4 °C for 18 h. The collagen content of the lungs was determined using the SircolTM Soluble Collagen Assay Kit (Biocolor Ltd., Carrickfergus, UK).

### 4.10. qRT-PCR Analysis of the Lung In Vivo

Total RNA was extracted from the lungs of the mice on day 28. The primer sequences for *TNF-α, IL-6, IL-1β, F4/80, MMP-9, TIMP-1,* and *GAPDH* (housekeeping gene) are summarized in Appendix A. The experiments were performed in triplicate.

### 4.11. Statistical Analysis

Statistical analyses were performed using the GraphPad Prism version 7.0 software (GraphPad, San Diego, CA, USA) and JMP^®^ 14 (SAS Institute Inc., Cary, NC, USA). Statistical significance was determined using a one-way analysis of variance with Tukey’s multiple comparison test and the Mann–Whitney test. Statistical significance was set at *p* < 0.05.

## 5. Conclusions

This study showed that hMIKO-1 significantly suppressed fibrosis in the lungs of BLM-ILD mice. Suppression of macrophage M2 polarization by hMIKO-1 is proposed as one of the mechanisms suppressing lung fibrosis in BLM-ILD mice. Further studies on the effects of hMIKO-1 on ILD are needed.

## Figures and Tables

**Figure 1 ijms-23-09669-f001:**
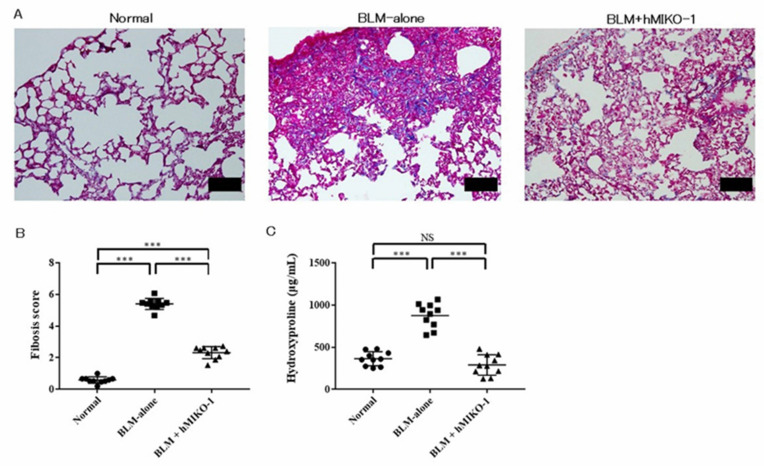
Human MIKO-1 reduces lung fibrosis in the lungs of mice 28 days after bleomycin administration. (**A**) Collagen deposition areas. Scale bars: 100 mm. (**B**) Fibrosis scores. (**C**) Collagen contents. Data are shown as mean ± SD (N = 10 mice per group). *** *p* < 0.001, significant differences between the linked groups. NS: not significant.

**Figure 2 ijms-23-09669-f002:**
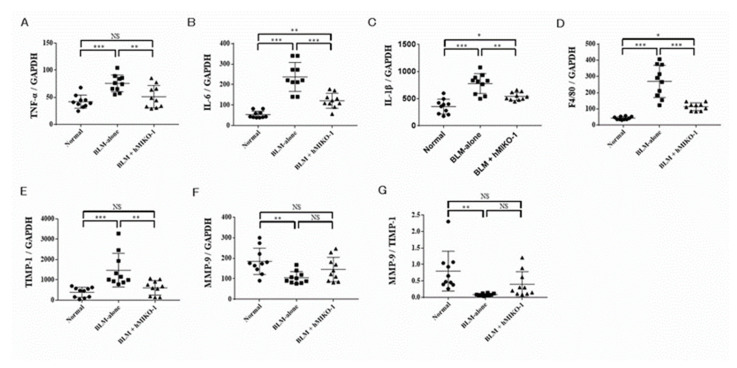
Quantitative reverse transcription-PCR analysis of lung mRNA expression. Relative mRNA expression of TNF-α (**A**), IL-6 (**B**), IL-1β (**C**), F4/80 (**D**), TIMP-1 (**E**), MMP-9 (**F**), and the MMP-9/TIMP-1 ratio (**G**) in the lungs of BLM-ILD mice on day 28. Data are shown as mean ± SD (N = 10 mice per group). * *p* < 0.05, ** *p* < 0.01, *** *p* < 0.001, significant differences between the linked groups. NS: not significant.

**Figure 3 ijms-23-09669-f003:**
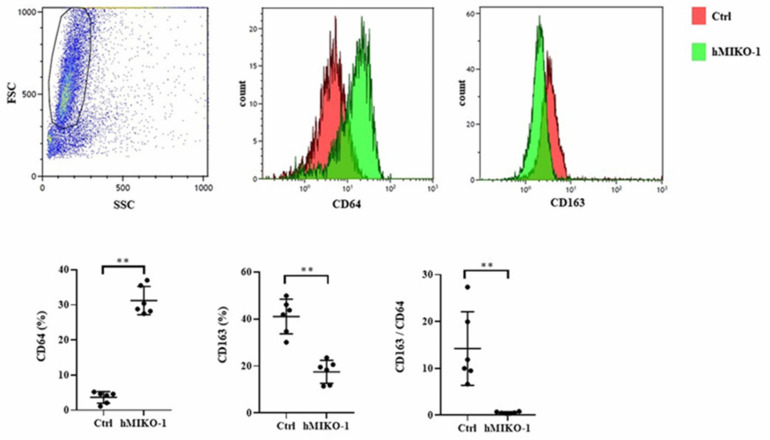
Cell surface antigens of murine macrophages. Expression of CD64 and CD163 and the CD163/CD64 ratio. hMIKO-1: hMIKO-1 co-culture group. Ctrl: Control group in which hMIKO-1 was not added. Data are shown as mean ± SD (N = 6 per group). ** *p* < 0.01, significant difference between the linked groups. NS: not significant.

**Figure 4 ijms-23-09669-f004:**
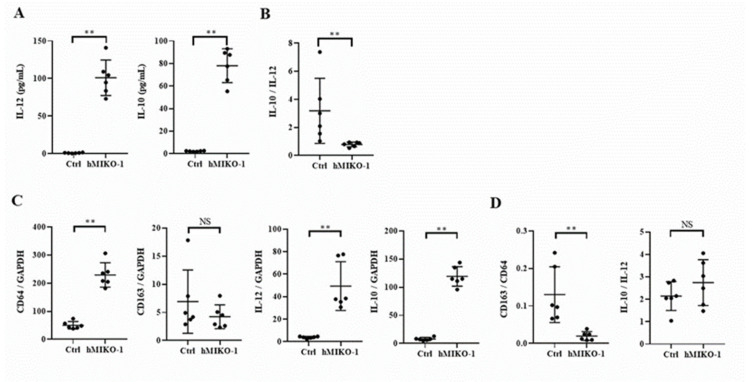
Cytokine concentration in the culture supernatant and gene expression in murine macrophages. Culture supernatant concentrations of IL-12 and IL-10 (**A**) and the IL-10/IL-12 ratio (**B**). Relative mRNA expression of CD64, CD163, IL-12, and IL-10 (**C**) and the CD163/CD64 and IL-10/IL-12 ratios (**D**) in murine macrophages. hMIKO-1: hMIKO-1 co-culture group. Ctrl: Control group in which hMIKO-1 was not added. Data are shown as mean ± SD (N = 6 per group). ** *p* < 0.01, significant differences between the linked groups. NS: not significant.

**Figure 5 ijms-23-09669-f005:**
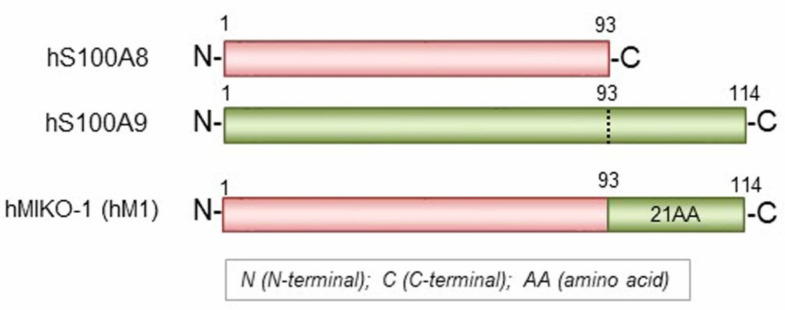
The whole frame of hMIKO-1. Full-length amino acid (AA) sequences of hS100A8 and hS100A9 are schematically represented by horizontal bars. The hMIKO-1 frame was as follows: 21 AA residues from the C-terminus of hS100A9 were added to the C-terminus of hS100A8.

## Data Availability

The data presented in this study are available on request from the corresponding author.

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
