# Peer review of "Human MIKO-1, a Hybrid Protein That Regulates Macrophage Function, Suppresses Lung Fibrosis in a Mouse Model of Bleomycin-Induced Interstitial Lung Disease"

_ijms, 2022, doi:10.3390/ijms23179669_

Round 1
Reviewer 1 Report
The authors report on the impact of hMIKO-1, a hybrid protein of S100A8 and S100A9, as an antifibrotic treatment of macrophage mediated lung fibrosis in response to bleomycin treatment of mice. The design is sound, the report is mostly clearly written (see below for word choices comments) although the report is fairly brief in each section. This is not a major problem, but there is room for a bit more explanation of results and possible implications. The novelty is the hMIKO-1 discovery and function highlighted in previous publications by members of this group, however this manuscript takes those observations to a direct test in vitro.
Minor concerns:
1. A few word choices or typos require one more editorial pass to best convey your meanings. Examples include; "and the inflammatory condition is poor" in line 33, poor seems a bit vague in this context, is there a better to describe the ILD inflammation level?; "irreversible lung fibrosis have become a social problem" in line 38, ILD syndromes affect society, but is that a social problem or a clinical problem in the context of this manuscript's focus?and "receptor-lie protein for rS100A8 with or without" is missing a k in line 55.
2. The manuscript is a bit more useful to others if a few interesting methods choices of wording are clarified. For instance, why "approximately 25 μg/mL" in line 220? Seems that was controlled and known. Is "culture supernatants were removed, and non-adherent" in line 221 accurate that non-adherent cells were stored for cytokine measures, or was the supernate, or combined supernate and non-adherent cell fraction saved? It matters as some cytokine secretion into the supernate likely occured, as implied in section 4.6. Also, "1,000 rpm" in line 224 is undoubtedly accurate and precise for your centrifuge, but what if mine has a different radius - tradition suggests using 'x g' force rather than rpm for this type of technical detail.
3. line 252 " the five most severely injured non-overlapping fields (magnification 200×) of 252 the lung parenchyma were evaluated" is an interesting way to sample tissue for your lung injury by Ashcroft scores. Do you have a reference for that sample method (using 5 worst segments as samples)? It implies you looked at the entire lung at 200x magnification. It is what you did and it can be justified, simply curious as to that choice.
Reviewer 2 Report
The authors present an interesting manuscript Human MIKO-1, a Hybrid Protein that Regulates Macrophage Function, Suppresses Lung Fibrosis in a Mouse Model of Bleo mycin-induced Interstitial Lung Disease , which is interesting, but some information is missing.
Why did you use only female and not male?
Dot plot graphs are better option.
I think that mean±SD is a better choice than mean±SEM.
